# Co-designing an intervention to increase uptake of advance care planning in later life following emergency hospitalisation: a research protocol using accelerated experience-based co-design (AEBCD) and the behaviour change wheel (BCW)

Anna-Maria Bielinska ![ORCID],[1,2] Stephanie Archer ![ORCID],[1,3] Ara Darzi,[1,2] Catherine Urch[1,2]

For numbered affiliations see end of article.

**Correspondence to**
Dr Anna-Maria Bielinska;
anna-maria.bielinska@doctors.org.uk

## ABSTRACT

**Introduction** Despite the potential benefits of advance care planning, uptake in older adults is low. In general, there is a lack of guidance as to how to initiate advance care planning conversations and encourage individuals to take action in planning their future care, including after emergency hospitalisation. Participatory action research methods are harnessed in health services research to design interventions that are relevant to end-users and stakeholders. This study aims to involve older persons, carers and healthcare professionals in co-designing an intervention to increase uptake of advance care planning in later life, which can be used by social contacts and healthcare professionals, particularly in the context of a recent emergency hospitalisation.

**Methods and analysis** The theory-driven participatory design research method integrates and adapts accelerated experience-based co-design with the behaviour change wheel, in the form of a collaborative multi-stakeholder co-design workshop. In total, 12 participants, comprising 4 lay persons aged 70+, 4 carers and 4 healthcare professionals with experience in elder care, will be recruited to participate in two online half-day sessions, together comprising one online workshop. There will be a maximum of two workshops. First, in the discovery phase, participants will reflect on findings from earlier qualitative research on views and experiences of advance care planning from three workstreams: patients, carers and healthcare professionals. Second, in the co-design phase, participants will explore practical mechanisms in which older persons aged 70+ can be encouraged to adopt advance care planning behaviours based on the behaviour change wheel, in order to co-design a behavioural intervention to increase uptake of advance care planning in older adults after an emergency hospitalisation.

**Ethics and dissemination** Ethical approval has been obtained from the Science Engineering Technology Research Ethics Committee at Imperial College London (Reference: 19IC5538). The findings from this study will

## STRENGTHS AND LIMITATIONS OF THIS STUDY

⇒ The study draws on previous research exploring the views of patients, carers and healthcare professionals in order to design a behavioural intervention.
⇒ Participants can reflect their own views and experiences during a highly collaborative co-design process, which recruits a diverse group of participants, including lay persons aged 70+, carers and healthcare professionals with experience in elder care.
⇒ To our knowledge this the first study to use the behaviour change wheel (BCW) to design a behavioural intervention to increase uptake of advance care planning in later life following an emergency hospitalisation.
⇒ To our knowledge this is also the first study combining accelerated experience based co-design with the BCW, other studies having used experience based co-design over longer time frames.
⇒ The online workshop might digitally exclude older adults with limited access to the internet.

be disseminated through publications, conferences and meetings.

## INTRODUCTION

Advance care planning (ACP) is a spectrum of approaches to planning future care[1] which 'supports adults at any age or stage of health in understanding and sharing their personal values, life goals and preferences'.[2] ACP can be started at any point in an individual's life course, including during chronic and serious illness.[3] It is an iterative process involving discussions between an individual and their care provider(s) and may cover a range of domains, from advance statements regarding care preferences, to advance decisions to

refuse treatment (ADRT). Individuals may also decide to nominate a surrogate decision-maker in case of future loss of capacity.[4] ACP has been highlighted as critical to supporting shared decision-making between patients and healthcare professionals[1] through maintaining the autonomy and self-determination of patients. Research in older adults supports early engagement in ACP and highlights its importance in current and future care, as this group may have multi-morbidities, complex psychosocial needs and face social isolation.[5–7]

ACP occurs in a range of settings including the home, general practice, nursing homes, specialist hospital outpatient services and hospital wards.[4] Traditionally, ACP was mainly pursued in palliative care,[8] however there is increased recognition that older hospitalised adults, particularly those who have had an emergency hospitalisation, are an important cohort for ACP.[1 9 10] As around one in five patients aged 70+ admitted to hospital as an emergency are in the last year of life,[1] acute hospitalisation can act as a trigger for tailored ACP.[1 10] ACP following hospitalisation is a dynamic process: preferences for life-sustaining treatment may change over the course of an acute illness or convalescence,[11] and the impact of uncertainty in later life on ACP is recognised by caregivers of acutely hospitalised older persons.[12] Qualitative research has shown that hospitalised older adults believe that ACP has a beneficial role in planning for their physical and psychosocial aspects of health, and patients look towards the collaborative expertise of clinicians to discuss and enact ACP.[1] Furthermore, older inpatients also regard ACP as having a role in contemplating the possibility of physical deterioration, death and dying when considering healthcare choices in later life.[1] Carers of older hospitalised adults regard ACP as process which helps to support older individuals, particularly in terms of well-being, quality of life and independence.[12]

Despite the potential benefits of ACP in older adults receiving emergency hospital care, such as improved concordance of care with patient's wishes, improved end-of-life care and better emotional well-being in relatives,[13] and notwithstanding the numerous care planning initiatives,[14 15] the uptake of ACP is low.[16 17] This is even among cohorts of patients who are facing life-changing illness, such as older and seriously unwell medical inpatients,[16] surgical trauma patients[18 19] and those attending cancer clinics.[17] As such, it is essential that the uptake of ACP is increased in all patient groups,[20] particularly older adults.[21] Several studies have focused on brief interventions delivered by clinicians and facilitators, empowering seriously ill older persons to recognise the importance of ACP in the emergency department,[22–24] with improved engagement in ACP and electronic documentation of healthcare proxy forms after emergency department visits.[23] While these interventions are extremely valuable in the emergency department, interventions to increase ACP that transcend other areas of health and or social care (eg. on the hospital ward or in the community) are also required.

## Rationale for the current study

Since ACP can be defined as a health behaviour, behaviour change models can help guide the development of interventions to increase the uptake of ACP.[21] Greater understanding of ACP as a health behaviour is important as criticism of ACP has highlighted the lack of consideration of the 'complexity, emotion and interpersonal elements of real-time decision-making', as limiting its uptake and impact.[18] Previous research has often limited ACP to completion of documentation or a one-off discussion with a clinician.[18] Brief interventions to increase ACP have been used with success in the emergency department setting by clinicians and ACP facilitators.[22–24] In contrast, our study examines ACP as behaviour to be explored within both a clinical and social context, using the event of an emergency hospitalisation as a pivotal 'teachable' event to reflect on care planning needs in the future. Co-design involving older persons, carers and clinicians addressing ACP behaviour post hospitalisation in this context is novel.

The Medical Research Council recommends that identifying and developing a theoretical understanding is integral to developing complex interventions, including understanding the process of change, rationale, expected impact and the mechanisms by which change is to be achieved.[25] While a wide range of behaviour change frameworks exist,[26] the behaviour change wheel (BCW) has been cited as a recommended resource when designing interventions.[27] The BCW is an evidence-based framework, which consists of a 'behavioural COM-B system' at the hub, encircled by 'intervention functions' and thereafter surrounded by 'policy categories'[26] (see figure 1).

The Capability, Opportunity, Motivation, Behaviour (COM-B) system is a framework that can be used to understand a behaviour and focuses on capability, opportunity and motivation.[26 27] There are nine intervention functions: education, persuasion, incentivisation, coercion, training, restriction, environmental restructuring, modelling and enablement.[26] The policy categories at the rim of the wheel support the delivery of the intervention.[26] The BCW has been used to guide the development of a range

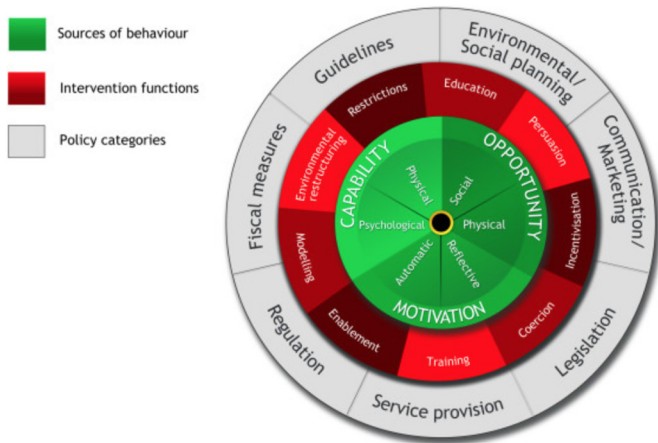

**Figure 1** The Behaviour Change Wheel (reproduced from[26]).

of interventions in both physical and mental health.[28 29] Specifically in older adults' healthcare, the BCW has been used in a variety of behavioural interventions, from improving exercise participation for older adults in the community,[30] to strengthening the role of nurses in stroke rehabilitation services,[31] altering medication prescribing practices for older adults[32] and preventing pressure ulcers in nursing homes.[33] Behavioural research by Sudore *et al* has used the Stages of Change model[34] to understand ACP as a health behaviour.[35] However, to date there are no published studies using the BCW to increase uptake of ACP in later life in any setting.

While the BCW may be an appropriate and effective theoretical model on which to base the 'active ingredients' of an intervention, other approaches for their design and delivery must be considered, to deliver an intervention that is acceptable to those who will use it and/or implement it in their services. One such method is experience based co-design (EBCD), which has been evaluated as an effective strategy for quality improvement in healthcare services through 'narrative-based, participatory action research' with patients, families and staff.[36] Recent robust theory driven approaches have combined EBCD and the BCW to guide complex intervention development co-design, involving service users and other stakeholders.[29] Both EBCD and the BCW are complementary and have been integrated in qualitative research.[29] The stepwise, reflective and pragmatic approaches of both EBCD and BCW help to provide solutions to benefit end-users in quality improvement initiatives.[26 36] EBCD involves six sequential steps (see figure 2).

While the original EBCD format can vary in the length of time required to complete the process, typically lasting 6–12 months and occasionally exceeding this,[37] a more expedited version is available.[36] In *accelerated* experience-based co-design (AEBCD), pre-existing collected narratives of experience from relevant stakeholders such as

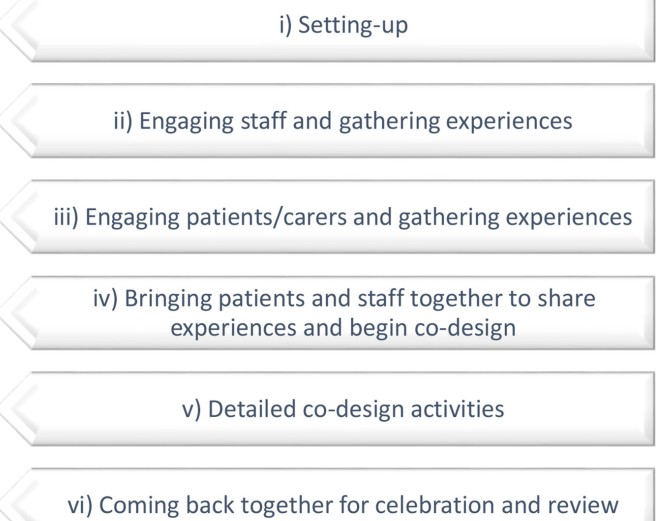

i) Setting-up

ii) Engaging staff and gathering experiences

iii) Engaging patients/carers and gathering experiences

iv) Bringing patients and staff together to share experiences and begin co-design

v) Detailed co-design activities

vi) Coming back together for celebration and review

**Figure 2** The steps of experience based co-design (adapted from[28 36]).

patients and staff form the initial data to start co-design while maintaining the 'rigorous and effective patient-centred quality improvement approach'.[36]

## Summary and aims

There is increasing evidence that an emergency admission may herald the last year of life in a subset of older adults[1 10]; increased uptake of ACP may be beneficial in this group. To address this need, the aim of this study is to use the BCW to co-design a behavioural intervention to increase uptake of ACP in later life after an emergency hospitalisation, through AEBCD with older persons, carers and healthcare professionals.

## METHODS AND ANALYSIS
### Study design

The research study will use AEBCD within an online workshop (split into two sessions) to co-design an intervention to increase the uptake of ACP in the over 70s. In line with AEBCD methodology,[36] the qualitative data collected from earlier studies (see box 1) will be used as a starting point.

The AEBCD workshop will be divided into two phases: an initial discovery phase and a second co-design phase. The discovery phase will present the previously collected data on ACP. Workshop participants will be asked to reflect and discuss any experiences and views of ACP. This phase of the workshop will allow the group to identify and develop potential improvements to increase uptake of ACP (see online supplemental file 1 session 1, timetable A).

The co-design phase will draw on the steps for intervention design using the BCW[28] (see figure 3).

While the BCW highlights the range of potential mechanisms which could be from the theoretical basis of our intervention, data from the prior qualitative research studies and wider literature has highlighted that increasing motivation via education may be efficacious in increasing uptake of ACP.[1 12 38] As such, this stage of the co-design workshop will focus on developing the context/setting, format and content of an educational resource. This will be achieved through prioritising important values and behaviours for ACP in later life, and will consider the **a**cceptability, **p**racticability, **e**ffectiveness, **a**ffordability, **s**ide-effects and e**q**uity (APEASE)[27] of any intervention (see online supplemental file 1 timetable B).

### Patient and public involvement

This is a participatory action approach study. The design and delivery of the qualitative research presented in the discovery phase of the workshop involved older hospitalised patients, carers and clinicians across medicine and surgery. The views and experiences of the public were consulted in preparation for this study. Local primary care practitioners provided feedback at an early stage of the study design to ensure its relevance across different care settings. The study concept, that is,

### Box 1   Prior work informing the design of this study

In preparation for this study, the experiences and views of older patients, carers and healthcare professionals of advance care planning (ACP) in later life had been collected via qualitative interviews.[1 12 45] Across three qualitative interview studies, a total of 48 participants took part—20 hospitalised older adults, 8 carers and 20 frontline healthcare professionals. Interviews with older adults aged 70+ admitted to hospital have shown that they believe that ACP is relevant to their care following an emergency hospitalisation across a spectrum of prognoses and that they look towards the expertise of healthcare professionals after a medical emergency to plan their future care collaboratively.[1] Carers of older adults who have been hospitalised believe that ACP has a role in supporting individuals in later life and that enabling the ACP process is crucial through practical support mechanisms.[12] Interviews with frontline healthcare professionals have shown that open conversations with patients regarding ACP are valued in guiding clinicians' decisions and that there is a role in educating the clinical workforce to better understand how to approach ACP (unpublished data).

designing an intervention to increase uptake of ACP, was discussed at an interdisciplinary stakeholder group with a specialist interest in ACP, including nursing and patient representatives.

### Study setting

The study will be conducted online on Microsoft Teams hosted by Imperial College.

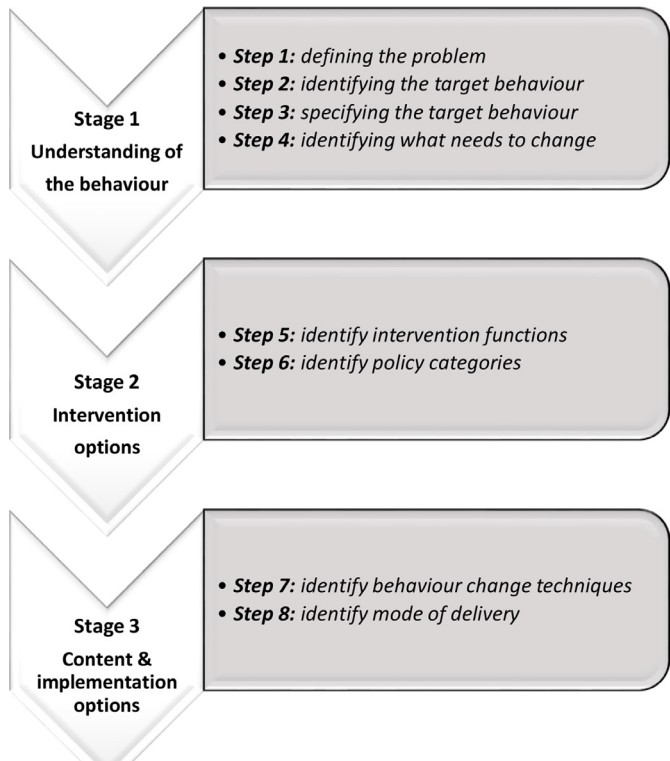

**Figure 3** Behaviour change wheel steps for intervention development (adapted from[28]).

### Study participants

The AEBCD workshop will comprise of approximately 12 participants as follows:

► Four lay persons with experience of being a patient aged 70+ (ideally who have experienced an emergency admission).
► Four lay persons with experience of being a carer of an older person (particularly for those aged 70+).
► Four health and social care professionals with experience of caring for older patients aged 70+.

We have decided to limit the number of participants per workshop to encourage in-depth discussions during a fixed-time period and encourage inter-participant and participant-facilitator interactions for productive co-design activities. However, there is scope for the workshop to be repeated, consistent with EBCD methodology, where multiple activities are conducted over time.[36]

### Sampling method

The study will be advertised to lay persons via the Imperial Research Partners Group, Patient and Public Involvement initiatives and third sector. Lay persons interested in participating can contact the study team for more information. Healthcare professionals will be invited from Imperial College and other academic institutions, including via email to existing contacts who have experience of caring for older persons. Those who wish to participate will be able to respond by email to the study team.

Co-design workshops may attract participants with a specific interest in the subject area. To help reduce the risk of recruitment bias, the online platform VOICE will be used to recruit lay persons. VOICE is a general platform for public involvement in research, rather than a specialist organisation for end-of-life research.[39] Furthermore, financial re-imbursement will be provided for lay participants, in line with guidance for public involvement by the National Institute for Health Research (NIHR) INVOLVE.[40]

Where possible, purposive sampling will be used to achieve a diverse group of older persons and carers, in terms of a range of lived experience of illness, caring and ACP, in addition to purposive sampling of multidisciplinary healthcare professionals across hospital and community settings. The VOICE platform for public involvement gathers information on equality and diversity, to help better understand the backgrounds of those involved in research, as per the recommendation of the NIHR. Any limitations in the diversity or inclusiveness of the sample will be acknowledged.

### Participant inclusion criteria

The participant inclusion criteria are as follows:

► Participants must be able and willing to participate in a group discussion.
► Participants must have experience of being a patient, caring or are a healthcare professional with experience of older person's healthcare.

► Lay persons with experience of being a patient must be aged 70+
► Carers and healthcare professionals must be aged 18+
► Participants must have capacity to give informed consent.
► Participants must be able to speak, read and write in English.

## Participant consent

Consent to enter the study will be sought from each participant after a full explanation has been given, a participant information sheet offered and time allowed for consideration (at least 24 hours). Signed participant consent will be obtained. The right of the participant to refuse to participate without giving reasons will be respected and all participants will be free to withdraw at any time. If participants have already taken part in the workshop, their contribution will not be withdrawn as it contributes to a wider discussion, but no extracts from their contributions within the discussions will be used in any write-up or presentation.

## Study procedures

### Terminology used

Some researchers use the term 'future care planning' instead of 'advance care planning' as this is more intuitively understood.[41] 'Future care planning' will be used in this study with participants to aid the flow of the discussion.

### Baseline questionnaire

Data will be individually gathered on basic demographic information such as age, sex and background (eg, whether they have experience of being a patient, carer or healthcare professional) through a brief questionnaire distributed via email. Within the questionnaire, participants will be asked if they have already, or plan in the future, to complete ACP (see online supplemental appendix).

### Evaluation

Participants will have an opportunity to debrief and feedback on the day of the workshop. They will also have email contact details if they wish to provide written feedback between or after the sessions.

### Follow-up

Following completion of the 1-day workshop, participants will be contacted once approximately 4 weeks after the AEBCD event, via telephone or email to ask whether they have or plan to complete an ACP in the future. This follow-up phone call would last a maximum of 5 min and does not involve creating an ACP. After the study, participants will be given the opportunity to receive updates regarding the study via email.

### Study duration

We plan to conduct a minimum of one and a maximum of two AEBCD workshops.

The workshop will be repeated if further data is required following the initial workshop to form an intervention or guidance to facilitate ACP post-hospitalisation from the perspective of all three stakeholders (older persons, carers and clinicians), or if there is a need to purposefully recruit additional participants to achieve a balance of older persons, ideally with varied lived experience, carers and multidisciplinary clinicians

Each online workshop lasts the course of 1 day and is split into two separate half-day sessions. Completion of the 1-day workshop should take approximately between 4 and 5 hours, including comfort breaks. Each participant would participate in only one workshop. The start date of the study is late summer of 2021.

### Data analysis

Responses from the participant questionnaire will be analysed to provide a descriptive summary of the participant characteristics. The information provided in Phase 1 (the discovery phase) will inform the co-design activities of session 2. On completion of the workshop, the audio recordings will be transcribed verbatim. The principal investigator will listen to the recordings and review the transcripts. Other researchers will only have access to the transcripts.

An inductive approach using thematic analysis[42] will be adopted. This 'bottom-up' approach to data analysis will allow us to explore elements related to the design of the intervention while more fully capturing the experiences of our participants that are not directly linked to intervention development. As a result, we will be able to use data from the study to inform intervention development (via the BCW/COM-B framework/Behaviour Change Technique Taxonomy (BCTT))[43] and provide valuable insights into lived experiences of ACP, which may be disseminated separately. The co-design process will be reported following guidance for reporting intervention development studies.[44]

We will use the findings of this study to generate a prototype educational intervention, which can be further refined and tested in future studies with a diverse group of participants. Other than a general reflection, developing specific detailed health policies is beyond the scope and time-constraints of the workshop—this would require future collaboration with the wider research team and other stakeholders relevant to the setting(s) where the policy may be implemented.

## ETHICS AND DISSEMINATION

### Ethical approval

Ethical approval has been obtained from the Science Engineering Technology Research Ethics Committee (SETREC) at Imperial College London (SETREC Reference: 19IC5538).

### Ethical considerations

Informed consent of participants is integral to the study as already outlined in the study methods. Ethical issues

relating to data management, participant well-being and digital exclusion are detailed below.

## Data management and monitoring

The principal investigator will preserve the confidentiality of participants taking part in the study and fulfil transparency requirements under the General Data Protection Regulation for health and care research. Data and all appropriate documentation will be confidentially and securely stored for a minimum of 10 years after the completion of the study, including the follow-up period.

## Consideration of well-being of participants

To respect participants' mental well-being, participants will be informed prior to the workshop that anyone can take a break whenever needed. Participants will be invited to speak with a mental health first aider after the meeting should the workshop raises any emotional concerns. To help participants feel at ease from the outset and reduce any power dynamics between professional clinicians and lay persons, there will be an informal icebreaker at the outset of the workshop. The facilitator will also highlight the value of both personal and professional experts throughout the workshop. If necessary, participants may access Jamboard (a virtual whiteboard via link in the chat function of Microsoft Teams). This will give attendees the option of writing personal/anonymous post-it notes to ensure that their comments are being captured.

## Minimising digital exclusion

To reduce the risk of digital exclusion, participants will have an opportunity to read software information before sessions, if needed, to feel more comfortable. Participants will also be able to access any software before the start of the session and ask questions. During the workshop, participants can write comments within Microsoft Teams sessions for any further IT support.

## Dissemination

In line with the final stage of the AEBCD method, the co-designed content will be 'celebrated' and showcased in patient, carer and clinician forums, initially within Imperial College, the host organisation. The findings from this study will also be disseminated through academic publications, conferences and meetings

**Author affiliations**
[1]Department of Surgery and Cancer, Imperial College London, London, UK
[2]Department of Surgery, Cancer and Cardiovascular, Imperial College Healthcare NHS Trust, London, UK
[3]Department of Public Health and Primary Care and Department of Psychology, University of Cambridge, Cambridge, UK

**Acknowledgements** Anna Lawrence-Jones, Patient and Public Involvement & Engagement Lead at the Institute of Global Health Innovation, Imperial College London, for advice on facilitating online workshops to encourage collaboration within the co-design participant team. Soh-yon Park, Design Researcher at the Institute of Global Health Innovation, Imperial College London, for expertise on facilitating workshop discussions.

**Contributors** A-MB conceptualised and designed the study protocol under supervision from SA and CU. A-MB wrote the initial protocol. SA and CU amended the initial protocol. A-MB, SA, AD and CU read, advised and approved of the protocol prior to submission.

**Funding** The Department of Surgery and Cancer receives infrastructural support by the Imperial NIHR Biomedical Research Centre (BRC). Funding granted via the National Institute for Health Research Patient Safety Translation Research Centre (NIHR PSTRC), grant reference number PSTRC-2016–004.

**Competing interests** None declared.

**Patient and public involvement** Patients and/or the public were involved in the design, or conduct, or reporting, or dissemination plans of this research. Refer to the Methods section for further details.

**Patient consent for publication** Not applicable.

**Provenance and peer review** Not commissioned; externally peer reviewed.

**ORCID iDs**
Anna-Maria Bielinska http://orcid.org/0000-0001-8668-771X
Stephanie Archer http://orcid.org/0000-0003-1349-7178

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
