## [Reviewer comments · BMJ Open]

ARTICLE DETAILS

TITLE (PROVISIONAL)	Co-designing an intervention to increase uptake of Advance Care Planning in later life following emergency hospitalisation: a research protocol using Accelerated Experience-Based Co-design (AEBCD) and The Behaviour Change Wheel (BCW).
AUTHORS	Bielinska, Anna-Maria; Archer, Stephanie; Darzi, Ara; Urch, Catherine

VERSION 1 – REVIEW

REVIEWER	Jones, K The Open University, HWSC WELS
REVIEW RETURNED	01-Sep-2021

GENERAL COMMENTS	Dear Author (s), Thank you for submitting this abstract for consideration for publication to the BMJ. This is a timely study addressing the pressing issue of low uptake among older adults in advance care planning. I concur that this study is needed given the lack of overall guidance. That this is a study which involved potential service users in co-production adds value to this study. My concern is the relatively low number of participants. Perhaps there is a justification for this and I would urge the authors to include this.
--

REVIEWER	Fromme, Erik Ariadne Labs
REVIEW RETURNED	14-Nov-2021

GENERAL COMMENTS	1.) This research protocol describes the design phase for a planned study, rather than the study itself. I am uncertain, but it seems like this is not what the Editors meant by "Protocol papers should report planned or ongoing studies." It seems like the planned study would be a better candidate for publication than the workshops that will inform the eventual study design. 2.) The main criticism I have of your method is one that has plagued advance care planning research. It is extremely difficult to recruit volunteers for ACP research who are not highly enthusiastic about ACP unless you pay them. You end up with an intervention designed by the people most likely to want to do it, whereas what you really want is input from people who are, at best, ambivalent about ACP. You don't need to design your intervention for people who are already enthusiastic, you need to design it for people who are ambivalent but might be swayed by an intervention designed specifically for them. However even a perfectly designed intervention will still fail if it's delivered by a not completely
---

	trustworthy person, at the wrong time, or explained in the wrong way. 3.) Given your interest in using acute hospitalization as a trigger, you should consider the work of Peter Ditto, who studied how ACP preferences change following hospitalization, then often revert if the patient is able to get better. See particularly: Ditto, P.H., J.A. Jacobson, W.D. Smucker, J.H. Danks, and A. Fagerlin, Context changes choices: a prospective study of the effects of hospitalization on life-sustaining treatment preferences. Med Decis Making, 2006. 26(4): p. 313-22.PMCID#:16855121. 4.) You've probably seen the recent editorials by R. Sean Morrison and others pointing out the lack of results in ACP research going back decades. I think it's worth including a rationale for why your (planned) study will be different.
--	---

REVIEWER	Sinclair, Craig University of New South Wales, School of Psychology
REVIEW RETURNED	22-Nov-2021

GENERAL COMMENTS	Thank you for the opportunity to review this manuscript. The authors describe an Accelerated Experience Based Co-Design process in a protocol paper for a participatory co-design study to inform an advance care planning intervention for older adults discharged from hospital after an emergency hospitalisation. I agree with the authors comments that this method of involving people with relevant lived experience in the design of advance care planning interventions addresses a gap in the research literature. I have some comments which I would like to see addressed.  1. The COM-B framework, in particular the intervention functions and policy categories, resonate for implementation and 'scaling up' of an intervention, but it is not clear how these are directly relevant to the co-design process and the lived experiences of the participants, in particular the older adults and/or carers. How will the authors respond if participant responses raise themes that do not map onto the COM-B framework? 2. The FCP acronym for future care planning is introduced at line 45 without definition. I would prefer that the authors avoid interchanging between ACP and FCP, but instead adopt one term. 3. The authors intend to undertake 1-2 workshops. How will this decision be determined? Is it based on participant availability/willingness? Or the collection of a necessary amount of information to inform the intervention? 4. There are some good measures in place to support participant emotional safety and avoid digital exclusion of participants. However the detailed processes of the AEBCD workshops are unclear, aside from the provision of a workshop timetable. How will the discussion be facilitated? How will the authors ensure that all voices are heard and considered? Is there a process for ensuring that participants are satisfied with the interpretation of their input and/or its implementation into the final intervention?
---

	5. Given the small number of representatives from each of the perspectives, it is important to consider the extent to which this group might be meaningfully able to comment on issues of diversity and inclusiveness for an intervention. Can any consideration be given to these issues in the sampling of participants?
--	--

REVIEWER	Ouchi, Kei Dana-Farber Cancer Institute
REVIEW RETURNED	22-Nov-2021

GENERAL COMMENTS	Thank you for the opportunity to review the protocol titled: Co-designing an intervention to increase uptake of Advance Care Planning in later life following emergency hospitalisation: a research protocol using Accelerated Experience-Based Co-design (AEBCD) and The Behaviour Change Wheel (BCW). This is an interesting approach to exploring and designing a behavioral intervention to increase advance care planning following emergency hospitalizations. The theoretical framework seems to be established, and writing is clear. I do not have any specific feedback about the protocol clarify. The only comment I have is that this type of behavioral intervention has been extensively studied by others. For example, Dr. Rebecca Sudore has done phenomenal work in this area already for seriously ill older adults: https://profiles.ucsf.edu/rebecca.sudore Specifically for seriously ill older adults after emergency department visits, I have similar behavioral interventions that I am working on: https://pubmed.ncbi.nlm.nih.gov/30418094/ https://pubmed.ncbi.nlm.nih.gov/32471321/ https://pubmed.ncbi.nlm.nih.gov/30223014/ I always welcome more studies to get seriously ill older adults to engage in focused, advance care planning conversations after acute health declines. So much remains to be unanswered. At the same time, I am a little worried about the duplicative nature of this work to what is in the literature. Other minor suggestions for improvement are:  1. Introduction is too long to articulate the necessary significance and innovation of the study. 2. Please consider adding qualitative framework to guide the discovery and intervention design phases of this study.
---

VERSION 1 – AUTHOR RESPONSE

Reviewer comments

Reviewer 1: Dr K Jones, The Open University - Comments to the Author

1. Dear Author (s), Thank you for submitting this abstract for consideration for publication to the BMJ. This is a timely study addressing the pressing issue of low uptake among older adults in advance care planning. I concur that this study is needed given the lack of overall guidance. That this is a study which involved potential service users in co-production adds value to this study.

Thank you for reviewing this study and for the positive feedback on our research in advance care planning (ACP).

2. My concern is the relatively low number of participants. Perhaps there is a justification for this and I would urge the authors to include this.

Thank you for this feedback. Although the protocol limits the number of participants to 12 per workshop, there is scope for the workshop to be repeated. This is consistent with the published literature on experience-based co-design, with multiple activities conducted over time (Locock, 2014). We are limiting the number of persons per workshop to encourage in-depth discussions during a fixed-time period and encourage inter-participant and participant-facilitator interactions for productive co-design activities.

The “study participants” section of the manuscript has therefore been updated as follows:

“We have decided to limit the number of participants per workshop to encourage in-depth discussions during a fixed-time period and encourage inter-participant and participant-facilitator interactions for productive co-design activities. However, there is scope for the workshop to be repeated, consistent with experience-based co-design methodology, where multiple activities are conducted over time (Locock, 2014)”.

Reviewer 2: Dr. Erik Fromme, Ariadne Labs - Comments to the Author

1. This research protocol describes the design phase for a planned study, rather than the study itself. I am uncertain, but it seems like this is not what the Editors meant by "Protocol papers should report planned or ongoing studies." It seems like the planned study would be a better candidate for publication than the workshops that will inform the eventual study design.

Thank you for raising this point. This is a protocol for an ongoing co-design study, a specific type of study within health service research. BMJ Open has previously published other protocols for high quality co-design studies, including those that focus on behavioural change (Brown, 2020; Law, 2020; Østervang, 2020). As such, our paper is in line with the publishing history of BMJ Open and meets their editorial requirements for protocol papers.

Brown MC, Araújo- Soares V, Skinner R, et al. Using qualitative and co- design methods to inform the development of an intervention to support and improve physical activity in childhood cancer survivors: a study protocol for BEing Active after ChildhOod caNcer (BEACON). *BMJ Open* 2020;10:e041073. doi:10.1136/ bmjopen-2020-041073

Law R-J, Williams L, Langley J, et al. 'Function First—Be Active, Stay Independent'—promoting physical activity and physical function in people with long- term conditions by primary care: a protocol for a realist synthesis with embedded co-production and co-design. *BMJ Open* 2020;10:e035686. doi:10.1136/ bmjopen-2019-035686

Østervang C, Lassen AT, Jensen CM, et al. How to improve emergency care to adults discharged within 24 hours? Acute Care planning in Emergency departments (The ACE study):a protocol of a participatory design study. *BMJ Open* 2020;10:e041743. doi:10.1136/ bmjopen-2020-041743

2. The main criticism I have of your method is one that has plagued advance care planning research. It is extremely difficult to recruit volunteers for ACP research who are not highly enthusiastic about ACP unless you pay them. You end up with an intervention designed by the people most likely to want to do it, whereas what you really want is input from people who are, at best, ambivalent about ACP. You don't need to design your intervention for people who are already enthusiastic, you need to design it for people who are ambivalent but might be swayed by an intervention designed specifically for them. However even a perfectly designed intervention will still fail if it's delivered by a not completely trustworthy person, at the wrong time, or explained in the wrong way.

Thank you for raising this important limitation. The manuscript now highlights that co-design workshops may recruit participants with a specific motivation in the work, potentially introducing bias. To examine this issue, we have included baseline and post-workshop screening questions to assess participants' readiness to make an advance care plan for themselves. To help mitigate the risk of recruitment bias, we have used the online platform VOICE for recruitment of lay persons in our co-design research - this is a general platform for public involvement in research and not a specialist organisation for end-of-life research (VOICE, 2022). Furthermore, since the time of manuscript submission, our research group has been granted funding via the National Institute for Health Research Patient Safety Translation Research Centre (NIHR PSTRC) at Imperial College London to provide payment for lay participants, in line with guidance for public involvement by NIHR INVOLVE (NIHR, 2021). We acknowledge that to assess the effectiveness of any co-designed intervention, a separate future study will be required.

The “sampling method” section of the manuscript has been updated to include relevant information, including bias and re-imburement:

“Co-design workshops may attract participants with a specific interest in the subject area. To help reduce the risk of recruitment bias, the online platform VOICE will be used to recruit lay persons. VOICE is a general platform for public involvement in research, rather than a specialist organisation for end-of-life research (VOICE, 2022). Furthermore, financial re-imburement will be provided for lay participants, in line with guidance for public involvement by the National Institute for Health Research (NIHR) INVOLVE (NIHR, 2021).”

The “data analysis” section of the manuscript has also been updated to include the following statement:

“We will use the findings of this study to generate a prototype educational intervention, which can be further refined and tested in future studies with a diverse group of participants.”

The funding statement has been updated to read as follows:

“Funding granted via the National Institute for Health Research Patient Safety Translation Research Centre (NIHR PSTRC) -grant reference number PSTRC-2016-004.”

3. Given your interest in using acute hospitalization as a trigger, you should consider the work of Peter Ditto, who studied how ACP preferences change following hospitalization, then often revert if the patient is able to get better. See particularly: Ditto, P.H., J.A. Jacobson, W.D. Smucker, J.H. Danks, and A. Fagerlin, Context changes choices: a prospective study of the effects of hospitalization on life-sustaining treatment preferences. *Med Decis Making*, 2006. 26(4): p. 313-22. PMID#:16855121.

Thank you for recommending Ditto’s research which examines the temporal changes in preference for life-sustaining treatment post-acute illness and hospitalisation. This relevant study has now been cited in introduction of the manuscript highlighting the dynamic nature of ACP changes post-hospitalisation.

The introduction of the manuscript has therefore been updated as follows:

“ACP following hospitalisation is a dynamic process: preferences for life-sustaining treatment may change over the course of an acute illness or convalescence (Ditto, 2006), and the impact of uncertainty in later life on ACP is recognised by caregivers of acutely hospitalised older persons (Bielinska, 2020).”

4. You've probably seen the recent editorials by R. Sean Morrison and others pointing out the lack of results in ACP research going back decades. I think it's worth including a rationale for why your (planned) study will be different.

Many thanks for this feedback and for highlighting the work of R. Sean Morrison - his recent editorial on the challenges and limitations of ACP in clinical practice and within research has now been cited in our updated manuscript.

The strength of our co-design study is that it involves older persons, carers and clinicians. To our knowledge, manuscripts detailing the processes of co-design involving all three stakeholder groups addressing ACP behaviour post hospitalisation have not yet been published. Previous research has often limited ACP to completion of documentation or a one-off discussion with a clinician. In contrast, our study examines ACP as behaviour to be explored within both a clinical and social context, using the event of an emergency hospitalisation as a pivotal "teachable" event to reflect on care planning needs in the future. It utilises the Behaviour Change Wheel to examine the psychological mechanisms underlying the initiation/engagement in ACP.

The introduction now references the Morrison editorial when mentioning the lack of uptake of ACP:

"...the uptake of ACP is low...This is even amongst cohorts of patients who are facing life-changing illness, such as older and seriously unwell medial inpatients (Waller, 2019), surgical trauma patients (Morrison, 2020; Yadav, 2017) and those attending cancer clinics (Waller, 2019)."

The "rationale for the current study" section of the manuscript has updated as follows:

"Since ACP can be defined as a health behaviour, behaviour change models can help guide the development of interventions to increase the uptake of ACP (Fried et al., 2012). Greater understanding of ACP as a health behaviour is important as criticism of ACP has highlighted the lack of consideration of the "complexity, emotion and interpersonal elements of real-time decision making", as limiting its uptake and impact (Morrison, 2020). Previous research has often limited ACP to completion of documentation or a one-off discussion with a clinician (Morrison, 2020). Brief interventions to increase ACP have been used with success in the emergency department setting by clinicians and ACP facilitators (Ouchi, 2019; Pajka, 2021; Leiter, 2018). In contrast, our study examines ACP as behaviour to be explored within both a clinical and social context, using the event of an emergency hospitalisation as a pivotal "teachable" event to reflect on care planning needs in the future. Co-design involving older persons, carers and clinicians addressing ACP behaviour post hospitalisation in this context is novel."

The use of the Behaviour Change Wheel has already been detailed in the "rationale for the current study" component of the manuscript.

Reviewer 3: Dr. Craig Sinclair, University of New South Wales -Comments to the Author

Thank you for the opportunity to review this manuscript. The authors describe an Accelerated Experience Based Co-Design process in a protocol paper for a participatory co-design study to inform an advance care planning intervention for older adults discharged from hospital after an emergency hospitalisation. I agree with the authors comments that this method of involving people with relevant lived experience in the design of advance care planning interventions addresses a gap in the research literature. I have some comments which I would like to see addressed.

1. The COM-B framework, in particular the intervention functions and policy categories, resonate for implementation and 'scaling up' of an intervention, but it is not clear how these are directly relevant to the co-design process and the lived experiences of the participants, in particular the older adults and/or carers. How will the authors respond if participant responses raise themes that do not map onto the COM-B framework?

Thank you for raising these points. We anticipate that the data obtained in the study will give insight into the sources of behaviour linked to the education intervention function we have already identified. Other than a brief reflection, we do not anticipate participants to identify or contribute to policy development as this is beyond the scope of the workshop. In order to develop policy around ACP, we plan to collaborate with the wider research team and other stakeholders relevant to the setting(s) where the policy may be implemented.

Responses from participants, including those which raise themes that cannot be mapped directly onto COM-B framework, may be used to develop guidance driven by lived experience on how to facilitate ACP post emergency hospitalisation in both social and clinical contexts. These responses may be collated and published separately, to support appropriate places for dissemination and impact

The data analysis component of manuscript has been updated to clarify these points:

“Other than a general reflection, developing specific detailed health policies is beyond the scope and time-constraints of the workshop - this would require future collaboration with the wider research team and other stakeholders relevant to the setting(s) where the policy may be implemented”.

“An inductive approach using thematic analysis (Braun and Clarke, 2006) will be adopted. This ‘bottom-up’ approach to data analysis will allow us to explore elements related to the design of the intervention whilst more fully capturing the experiences of our participants that are not directly linked to intervention development. As a result, we will be able to use data from the study to inform intervention development (via the BCW/COM-B framework/BCTT) (Michie, 2013) and provide valuable insights into lived experiences of ACP, which may be disseminated separately.”

2. The FCP acronym for future care planning is introduced at line 45 without definition. I would prefer that the authors avoid interchanging between ACP and FCP, but instead adopt one term.

Many thanks for noting this. The manuscript has been updated to use the ACP acronym throughout, including in the supplementary material. The FCP acronym has been deleted. The statement on “terminology used” within the study procedures section of the manuscript has been retained to maintain clarity on language.

3. The authors intend to undertake 1-2 workshops. How will this decision be determined? Is it based on participant availability/willingness? Or the collection of a necessary amount of information to inform the intervention?

Thank you for this question. The number of workshops will be determined by whether there is sufficient data collected from the initial workshop to form an intervention or guidance to facilitate ACP in this setting from the perspective of all three stakeholders (older persons, carers and clinicians). A further consideration will be whether each workshop has been able to achieve an appropriate balance of older persons, ideally with relevant and varied lived experience, carers and multidisciplinary health professionals.

The “study duration” section of the manuscript has been updated to clarify this point:

“The workshop will be repeated if further data is required following the initial workshop to form an intervention or guidance to facilitate ACP post-hospitalisation from the perspective of all three stakeholders (older persons, carers and clinicians), or if there is a need to purposefully recruit additional participants to achieve a balance of older persons, ideally with varied lived experience, carers and multidisciplinary clinicians.”

4. There are some good measures in place to support participant emotional safety and avoid digital exclusion of participants. However, the detailed processes of the AEBCD workshops are unclear, aside from the provision of a workshop timetable. How will the discussion be facilitated? How will the authors ensure that all voices are heard and considered? Is there a process for ensuring that participants are satisfied with the interpretation of their input and/or its implementation into the final intervention?

Thank you for the positive feedback and for your questions.

The discussion will be facilitated by a main facilitator presenting the content of previous research and asking participants questions about their experiences and views, with an emphasis on open questioning, asking probing questions where appropriate and clarifying participants’ comments. The main facilitator will also answer any other verbal questions that participants may have during the workshop, particularly related to ACP and behaviour change. In order to focus on presentation of content for the discovery phase and driving the verbal discussion during the workshop, the main facilitator will be assisted by two assistant facilitators for technical support, note-taking and timekeeping. The first assistant facilitator will take notes, attend to any technical software questions, and field text group chat questions in the Microsoft Teams interface, with the support of the second facilitator on this latter task if necessary. The second assistant facilitator will focus on time-keeping of the workshop, and ensuring all voices are heard and considered, including keeping track on which participants have spoken, which participants wish to further contribute, including noting the order of participants “raising hands” on Microsoft Teams.

There is a need to recognise within the facilitator and co-design team that an element of flexibility in the schedule may be required on the day to respond to participants' needs and contributions during a live workshop.

After the study, participants will be given the opportunity to receive updates regarding the study via email. Participants can provide additional feedback during and after the workshop if they wish - this is already outlined in the "evaluation" section of the manuscript.

The supplementary material regarding the running of the workshop has been updated to include the following:

"Recognition within the facilitator and co-design team that an element of flexibility in the schedule may be required on the day to respond to participants' needs and contributions during the live workshop."

The workshop outline in the supplementary material has therefore been updated to clarify the role of the workshop facilitators as follows:

"Persons involved in both sessions"

- Principal investigator (to act as main facilitator)
- Co-investigators (to act as assistant facilitators)
- Participants: 12 in total (consisting of 4 older persons, 4 carers and 4 healthcare professionals). An extra healthcare professional may need be invited in case of dropouts on the day of the workshop to balance the number of participants.

Role designation of workshop team to facilitate discussion

Role of the main facilitator

The discussion will be facilitated by a main facilitator presenting the content of previous research and asking participants questions about their experiences and views, with an emphasis on open questioning, asking probing questions where appropriate and clarifying participants' comments.

The main facilitator will also answer any other verbal questions that participants may have during the workshop, particularly related to ACP and behaviour change.

Role of the assistant facilitators

In order to focus on presentation of content for the discovery phase and driving the verbal discussion during the workshop, the main facilitator will be assisted by two assistant facilitators for technical support, note-taking and timekeeping.

The first assistant facilitator will take notes, attend to any technical software questions, and field text group chat questions in the Microsoft Teams interface, with the support of the second facilitator on this latter task if necessary.

The second assistant facilitator will focus on time-keeping of the workshop, and ensuring all voices are heard and considered, including keeping track of which participants have spoken, which participants wish to further contribute, including noting the order of participants “raising hands” on Microsoft Teams.

The following statement has also been added to the “follow-up” section of the manuscript:

“After the study, participants will be given the opportunity to receive updates regarding the study via email.”

5. Given the small number of representatives from each of the perspectives, it is important to consider the extent to which this group might be meaningfully able to comment on issues of diversity and inclusiveness for an intervention. Can any consideration be given to these issues in the sampling of participants?

Thank you for raising this point. Purposeful sampling will be used in the study to recruit a range of lay older person and carer participants, balancing relevant experiences of illness and caring where possible, in addition to multidisciplinary healthcare professionals across community and hospital settings. The online platform used to recruit lay participants (VOICE) gathers information on equality and diversity (VOICE, 2022), to help better understand the backgrounds of those involved in research, as per the recommendation of the NIHR.

To increase the diversity of our participants, there is the possibility to repeat the workshop targeting any specific groups that may have been under-represented in the initial workshop (this is mentioned in the updated “study duration” section of the manuscript). Whilst we will do our best to reach and select a diverse group of participants, the limitations of the diversity of the sample will need to be acknowledged. Any intervention developed could be piloted and refined over several rounds – some of which could be refined with specific groups if needed.

The “sampling method” section of the manuscript has therefore been updated as follows:

“Where possible, purposive sampling will be used to achieve a diverse group of older persons and carers, in terms of a range of lived experience of illness, caring and ACP, in addition to purposive sampling of multi-disciplinary healthcare professionals across hospital and community settings. The VOICE platform for public involvement gathers information on equality and diversity, to help better understand the backgrounds of those involved in research, as per the recommendation of the NIHR. Any limitations in the diversity or inclusiveness of the sample will be acknowledged.”

The following statement has been added to the “data analysis” section of the manuscript:

“We will use the findings of this study to generate a prototype educational intervention, which can be further refined and tested in future studies with a diverse group of participants.”

Reviewer 4: Dr. Kei Ouchi, Dana-Farber Cancer Institute -Comments to the Author:

1. Thank you for the opportunity to review the protocol titled: Co-designing an intervention to increase uptake of Advance Care Planning in later life following emergency hospitalisation: a research protocol using Accelerated Experience-Based Co-design (AEBCD) and The Behaviour Change Wheel (BCW). This is an interesting approach to exploring and designing a behavioral intervention to increase advance care planning following emergency hospitalizations. The theoretical framework seems to be established, and writing is clear. I do not have any specific feedback about the protocol clarify.

Thank you for reviewing this study and for the positive feedback on our research protocol in ACP.

2. The only comment I have is that this type of behavioral intervention has been extensively studied by others. For example, Dr. Rebecca Sudore has done phenomenal work in this area already for seriously ill older adults:

<https://profiles.ucsf.edu/rebecca.sudore>

Specifically for seriously ill older adults after emergency department visits, I have similar behavioral interventions that I am working on:

<https://pubmed.ncbi.nlm.nih.gov/30418094/>

<https://pubmed.ncbi.nlm.nih.gov/32471321/>

<https://pubmed.ncbi.nlm.nih.gov/30223014/>

I always welcome more studies to get seriously ill older adults to engage in focused, advance care planning conversations after acute health declines. So much remains to be unanswered. At the same time, I am a little worried about the duplicative nature of this work to what is in the literature.

Many thanks for your feedback and for welcoming our research. Thank you also for noting the existing literature in this key area. We had previously referenced research co-authored by

Sudore et al in our original manuscript (Sudore, 2017; Austin; 2015; Lum, 2015). In our revised manuscript, we have now cited the valuable studies from your research group, where brief interventions delivered by clinicians and facilitators have been used to empower seriously ill older adults to recognise the importance of ACP in the emergency department (Ouchi, 2019; Pajka, 2021; Leiter, 2018). We have also cited specific behavioural research by Sudore et al. using the Stages of Change model (Prochaska & DiClemente, 1983) to understand ACP behaviour (Sudore, 2014).

Our study provides a valuable and different contribution to research on ACP post emergency hospitalisation since our co-design protocol includes facilitating decision-making beyond the emergency department (e.g. inpatient wards and the community setting after hospital discharge). Our study encompasses long term planning, where appropriate, involving health and social care professionals following an emergency hospital stay. We have published research suggesting that older persons need support in facilitating ACP post emergency hospitalisation (Bielinska, 2021), therefore there is a need for research in our context using the lived experience of older persons, carers and clinicians.

Furthermore, our study is methodologically original, uniquely utilising AEBCD and the BCW to examine how to facilitate ACP. Sudore has used the Stages of Change model (Prochaska & DiClemente, 1983) to examine ACP (Sudore, 2014), rather than the Behaviour Change Wheel (BCW). We feel it is valuable for our research to be published to help inform the wider ACP research community, since they may wish to replicate or adapt the original design of our workshop using the Behaviour Change Wheel according to their needs.

The introduction section of the manuscript has therefore been updated to include the following:

“Several studies have focused on brief interventions delivered by clinicians and facilitators, empowering seriously ill older persons to recognise the importance of ACP in the emergency department (Ouchi, 2019; Pajka, 2021; Leiter, 2018), with improved engagement in ACP and electronic documentation of health care proxy forms after emergency department visits (Pajka, 2021). Whilst these interventions are extremely valuable in the emergency department, interventions to increase ACP that transcend other areas of health and or social care (e.g. on the hospital ward or in the community) are also required.”

“Behavioural research by Sudore et al. has used the Stages of Change model (Prochaska & DiClemente, 1983) to understand ACP as a health behaviour (Sudore, 2014). However, to date there are no published studies using the BCW to increase uptake of ACP in later life in any setting.”

“Greater understanding of ACP as a health behaviour is important as criticism of ACP has highlighted the lack of consideration of the “complexity, emotion and interpersonal elements of real-time decision making”, as limiting its uptake and impact (Morrison, 2020). Previous research has often limited ACP to completion of documentation or a one-off discussion with a clinician (Morrison, 2020). Brief interventions to increase ACP have been used with success in the emergency department setting by clinicians and ACP facilitators (Ouchi, 2019; Pajka, 2021; Leiter, 2018). In contrast, our study examines ACP as behaviour to be explored within both a clinical and social context, using the event of an emergency hospitalisation as a pivotal “teachable” event to reflect on care planning needs in the future. Co-design involving older persons, carers and clinicians addressing ACP behaviour post hospitalisation in this context is novel.”

Other minor suggestions for improvement are:

1. Introduction is too long to articulate the necessary significance and innovation of the study.

Thank you for this feedback. We have written the introduction for the benefit of the wider research community without specialist knowledge of ACP.

2. Please consider adding qualitative framework to guide the discovery and intervention design phases of this study.

Thank you raising this issue. We have carefully considered how best to analyse the data from the workshop. Whilst a deductive framework analysis using the Behaviour Change Wheel as a starting point could be used to guide interpretation of responses, we favour an inductive approach using thematic analysis (Braun and Clarke, 2006) mapping onto the Behaviour Change Wheel where appropriate. We prefer this approach as it allows us to more fully explore the lived experiences of participants that are not directly linked to intervention development (via COM-B framework), providing valuable insights into lived experiences of ACP.

The manuscript has therefore been updated as follows:

“An inductive approach using thematic analysis (Braun and Clarke, 2006) will be adopted. This ‘bottom-up’ approach to data analysis will allow us to explore elements related to the design of the intervention whilst more fully capturing the experiences of our participants that are not directly linked to intervention development. As a result, we will be able to use data from the study to inform intervention development (via the BCW/COM-B framework/BCTT) (Michie, 2013) and provide valuable insights into lived experiences of ACP, which may be disseminated separately.”

Reviewer: 1

Competing interests of Reviewer: None

Reviewer: 2

Competing interests of Reviewer: None

Reviewer: 3

Competing interests of Reviewer: I have no competing interests to report

Reviewer: 4

Competing interests of Reviewer: My work is similar to the study described.

VERSION 2 – REVIEW

REVIEWER	Jones, K The Open University, HWSC WELS
REVIEW RETURNED	12-Jan-2022

GENERAL COMMENTS	Thank you for your consideration of the comments from reviewing your paper on your proposed research. You have given due consideration and made changes which add strength and which I recommend for publication.
---

REVIEWER	Sinclair, Craig University of New South Wales, School of Psychology
REVIEW RETURNED	26-Jan-2022

GENERAL COMMENTS	The authors have satisfactorily addressed the Reviewers' comments in my view. I have no further suggestions for change.
---